Bots in software engineering: a systematic mapping study

Santhanam Sivasurya sivasurya.santhanam@dlr.de 1
Hecking Tobias 1
Schreiber Andreas 1
Wagner Stefan 2
1 Institute for Software Technology, German Aerospace Center (DLR) , Cologne , North Rhine-Westphalia , Germany
2 Institute of Software Engineering, University of Stuttgart , Stuttgart , Baden-Württemberg , Germany
Moparthi Nageswara Rao
Electronic publication date: 2022 Feb 9
Publication date: 2022
Volume: 8
Electronic Location ID: e866
Received 2021 Sep 8; Accepted 2022 Jan 5
Copyright: ©2022 Santhanam et al.
Copyright year: 2022
Copyright holder: Santhanam et al.
License: This is an open access article distributed under the terms of the Creative Commons Attribution License, which permits unrestricted use, distribution, reproduction and adaptation in any medium and for any purpose provided that it is properly attributed. For attribution, the original author(s), title, publication source (PeerJ Computer Science) and either DOI or URL of the article must be cited.
License URL: https://creativecommons.org/licenses/by/4.0/

Keywords: Bots, Software engineering, BotSE, Conversational AI, Digital assistants

Funding: The authors received no funding for this work.

==============================
Bots have emerged from research prototypes to deployable systems due to the recent developments in machine learning, natural language processing and understanding techniques. In software engineering, bots range from simple automated scripts to decision-making autonomous systems. The spectrum of applications of bots in software engineering is so wide and diverse, that a comprehensive overview and categorization of such bots is needed. Existing works considered selective bots to be analyzed and failed to provide the overall picture. Hence it is significant to categorize bots in software engineering through analyzing why, what and how the bots are applied in software engineering. We approach the problem with a systematic mapping study based on the research articles published in this topic. This study focuses on classification of bots used in software engineering, the various dimensions of the characteristics, the more frequently researched area, potential research spaces to be explored and the perception of bots in the developer community. This study aims to provide an introduction and a broad overview of bots used in software engineering. Discussions of the feedback and results from several studies provide interesting insights and prospective future directions.

Introduction

The process of software engineering becomes increasingly automated with the advancement of artificial intelligence techniques. One important strand in this regard are software development bots that act as (semi-)autonomous agents taking over common software engineering tasks, such as repository management or bug tracking. The term Bots—a short form for Software Robots—represents software programs from automated scripts to autonomous agents that execute actions when defined conditions are met. Many software developers, teams, and companies use bots to automate various—often repetitive—tasks to increase their work efficiency (Farooq & Grudin, 2016; Monperrus, 2019; Lebeuf, Storey & Zagalsky, 2017).

Bots that are designed to aid developers in the software development process in general are termed as Devbots. The developers can instruct the bots on what has to be done and with their pre-written logic, the bots help in executing the tasks. Devbots bridge the gap between human software development and automated processes (Van Tonder & Le Goues, 2019). In regulating software repositories, bots play the role of automated scripts for handling issues, managing PR, and CI pipelines. Apart from automating the recurring tasks, bots are also designed to access information through high level interfaces. Bots used for Question-Answering systems employ information retrieval methods. Bots that are accompanied with a goal for supporting developers to mitigate the workload are called digital assistants. To let the developer work without distraction or to avoid low level jobs, the digital assistant keeps track of tiny tasks, while the developer remains focused on the hard part of the problems (Murphy, 2019). Digital assistants consist of a wide range of systems from mobile apps to virtual characters in augmented reality. Among them, notable are the ones with conversational capabilities as they allow the developer to communicate with the system through Natural language commands. Researchers predict that conversational chatbot interfaces could revolutionize human–computer interaction (HCI) (Følstad & Brandtz, 2017). Bots and conversational UIs are even called “universal UI” and are seen as a paradigm shift for user interaction as they are natural for humans. The collaboration of humans with bots led to the term “Bot-ecosystems”, a place where humans and bots cooperate (Storey & Zagalsky, 2016).

Problem statement

Bots in software engineering come in diverse forms to be applied in different areas. They function as apps in maintaining repositories, as chat interfaces in question-answering systems, as digital assistants to assist developers with their tasks. Considering such a diverse spectrum of applications and varied functionalities, an overview of bots in software engineering has to be explored. This overview helps the researchers and developers to comprehend the existing territories of bots in software engineering and also to find out the unexplored potential research areas. Existing works on classification and selective analysis of bots provide an insufficient overall picture of the scope of bots. A study based on the published research articles on this topic will address this issue. The main problem addressed in this work is the categorization of bots in software engineering. We explored the various dimensions of such bots by attempting to answer what, why and how are bots used in software engineering.

Objective

We applied a systematic mapping study (SMS) process to study the span of bots in software engineering. The SMS outlines the explored areas of bots and the research hotspots. This leads us to discover available research gaps. In an interdisciplinary topic, the consolidated view on results and feedbacks from several articles help the community to evaluate bots from design, development, and maintenance perspectives. The research published in the past years have been collected, classified and analysed for a systematic study. The SMS provides a brief discussion on the application areas, the modes of interaction of such bots and also how they are developed considering design features. As a result of the study, valuable insights are inferred. There is lack of research towards voice controlled bots for software developers. For effective use of bots, design features must be considered as a prime aspect during bot development. In general, users accept bots in their development community with a few remarks such as noises, understandability, trust and explainability issues. We also discovered potential research ideas in this space.

Contribution

The contributions of this work are:

• A systematic mapping study of bots used in software engineering.

• Faceted classification of bots in software engineering.

• Existing software engineering areas which are exploited by the usage of bots.

• Identification of research gaps.

Outline

‘Background’ supplies a brief background information on bots. ‘Related Work’ lists out the related work done in this topic of research. ‘Search Criteria’ shows the search criteria used to filter out the literature for the mapping study. ‘Systematic Mapping Process’ lists out the parts in systematic mapping process. ‘Results’ discusses the classification of bots in detail and the existing work done in each category. ‘Discussion’ provides a brief discussion of the notable points from the results. ‘Research Gaps’ identifies the gaps in research on bots in software engineering. ‘Conclusions’ summarizes the work with conclusion and future works.

Background

The study done in this work mentions several bots used for various functionalities. In this section, we will provide a brief description about the addressed bots. Since the existing works classified bots under different dimensions, the categories are not always mutually exclusive and often overlap over one another. The terminologies are either declared based on the purpose and functionality of a bot, or it could represent a generic category that encompasses several bots. Codebots (Storey & Zagalsky, 2016) refer to bots that help make coding activities more efficient and effective. They are generic and could include all the bots which contribute towards improving coding tasks. Devops bots (Storey & Zagalsky, 2016) are those which are used to speed up code deployment, manage development infrastructure and operations. Documentation bots (Peng et al., 2018a) take care of writing natural language documentation for undocumented or poorly documented code.

Bots which are used in the testing environment are mentioned as testbots (Storey & Zagalsky, 2016). They run automated tests and check for errors. Bots are used not only to test for build fails, but when such a fail occurs, they even repair them. Repair bots (Urli, 2018; Monperrus et al., 2019) are built to repair build failures. By either proposing the changes via pull requests or suggesting the changes, these bots repair the builds automatically. A significant proportion of the bots in software engineering belong to managing repositories by maintaining issues, pull requests and continuous integration tools. Stalebots (Wessel et al., 2019) take care of outdated issues and pull requests. When issues and pull requests are idle for a certain period of time, stalebots close them.

As a collaborative platform, Github provides Github bots as a platform to develop and deploy bots in repositories. To automate the process of finding reviewers, Automatic reviewer recommendation bots (Peng et al., 2018b) suggest suitable reviewers to review the pull requests. Bots are even used outside the development environment. Support bots (Storey & Zagalsky, 2016) are those which are used as a communication medium between users and developers.

In human computer interaction, conversational user interfaces changed the input methods from the normal usage of physical hardware like mouse and keyboard devices to speech-based inputs. Such interfaces, when coupled with bots, are called conversational AI or conversational bots (Bradley, Fritz & Holmes, 2018; da Silva et al., 2020). When these conversational AI are employed in software engineering tasks, the developers can access and command bots through natural language communication. This establishes a natural form of communication between bots and developers and also reduces the learning curve.

This fundamentally changes the way we do software engineering. Bots armed with conversational UIs might constitute an important part of the software development process in the future. Interaction with bots through natural languages means that humans could interact with fellow developers and bots in the same mode of communication. With bots being visualized as a part of the developer community, conversational UI possess the necessary potential to make it a reality. Conversational bots range from chat-based communications to voice-controlled bots. Chatbots are the ones which restrict themselves with text conversations, whereas voice-assisted bots have added capabilities of speech interfaces. Speech interfaces are powered by speech recognition systems. These automatic speech recognition systems recognize the speech utterances made by the user and convert such speech into text. Chatbots are mainly used in communication platforms either among several developers or they serve as an interface to communicate with information sources.

Designing a bot involves setting a suitable persona for the bot. These personas are designated to bots depending upon the level of smartness they have been designed with. Chatbots(charlie) are such tools that communicate with the developer through a natural language interface, whereas autonomous bots(alex) indicate bots that work on their own i.e, the decision-making controls lie within the framework of the bot, and smart bot(sam) points to those bots which trigger events indirectly and are often trained using machine learning methods.

Related work

To study bots, it is not only important to know what is called a bot, but also what does not come under the description of bots. A recent report on the Dagstuhl seminar on bots in software engineering (Storey et al., 2020) provided a detailed discussion on this topic. In software engineering, bots are relatively new and ensuring the definition and distinction of terms used in this field is of great importance. Shihab, Wagner & Aurélio Gerosa (2021) provided a report on the Second International Workshop on Bots in Software Engineering (BotSE 2020); held in conjunction with the 42nd ACM/IEEE International Conference on Software Engineering (ICSE). Papers presented in the workshop are classified under three categories: Bots helping software development, Bot or not, and Bot recommendations & challenges. The community has significantly matured from simply focusing on the definition of what is a bot and the development of bots to performing specific tasks to focusing on more complex issues related to the effectiveness of bots (Shihab, Wagner & Aurélio Gerosa, 2021).

The list of previous work that contributed towards categorization and classification of bots in software engineering is limited. This is because bots are of various dimensions, and presenting a multidimensional view of bots in a specific field is not an easy task. Nevertheless, previous work contributed towards establishing a standard taxonomy for bots.

In an attempt to classify bots, Lebeuf et al. (2019) proposed a multi-faceted taxonomy in contrast to a hierarchical one so that the bots are labelled according to multidimensional categories. The taxonomy is extensive, but generic software bots are considered in this work. Erlenhov et al. (2019) tried a similar facet-based taxonomy on devbots by classifying bots based on Purpose, Initiation, Communication, and Intelligence. The work does not employ an extensive study, rather a limited number of devbots are considered in the work and the other dimensions such as the application area of the bots and design principles are not discussed in detail. From the taxonomy of contemporary devbots, prospective views of ideal devbots are sketched. As stated, the work was aimed at being a starting point for constructing an initial, extensible taxonomy. Storey & Zagalsky (2016) listed the types of bots used in software development based on the roles played by them as code bots, test bots, DevOps bots, support bots, and documentation bots and also how they improve the productivity of the developers. The other dimensions such as the persona of the bots and the nature of interactions are not discussed.

Eventhough conversational AI fits the description of bots, it has not been significantly considered in any of the previous works on bot classification. Since productivity in software engineering has been highly correlated with bots recently (Abdellatif, Badran & Shihab, 2016), categorization of bots in software engineering helps us to plot the existing work in the wide spectrum. Existing work on classification failed to provide an overall picture of this multidimensional view. In this work, we try to find the answer for the research question, how are bots classified in the software engineering considering their various dimensions. We attempted to categorize the varieties and subgroups of bots through responding to three sub-questions; why are bots used in software engineering, what types of bots are used there, how are the bots designed. We employed a systematic mapping study to help us find the answer to these three questions. The main focus of this work is to write a systematic mapping study on Bots used in software engineering, including current advancements and potential research ideas for the future.

Search Criteria

For the systematic mapping study, We defined the search terms considering the primary focus of the work. The following search terms are executed in the Google Scholar (https://scholar.google.com) database.

• “bots in software engineering”

• “conversational bots” AND “software engineering”

• “conversational AI” AND “software engineering”

The first term was obvious, as we had to focus “bots” that are used in “software engineering”. Since “conversational bots” and “conversational AI” represented the bots with natural language capabilities, they had to be included. Mere use of the term “conversational AI/bots in software engineering” lead to very few results since they indicated a very narrow domain and they are not as commonly used as “bots in software engineering”, but the use of AND operator between those two terms lead to results where the application or development of conversational bots/AI occurred in software engineering domain.

Few more restrictions are applied to the above search for a more specific collection for the study. The literature has to be published in English language and must be published in a time period from 2000–2021. These search terms are scanned for full-text search and the results returned are collected. We have also found that the BotSE repository (http://papers.botse.org) (Abdellatif, Badran & Shihab, 2016) is another excellent source to obtain bot related research. In total, the searches have returned 174 items.

We applied inclusion/exclusion criteria to restrict the irrelevant articles. Such filter is also followed by Petersen et al. (2008). These criteria help to obtain relevant research works in the focused area by setting boundaries. They also ensure reliable and reproducible results. The inclusion/exclusion criteria are as follows:

Inclusion criteria

1. Article must be about bots used in software engineering

2. Bots could be of various types from conversational bots, virtual assistants, development bots, repair bots to test bots

3. Tools which depict automatic or semi-automatic workflow execution that helps the developer’s work are also considered as bots.

4. Must contain a reasonable amount of material (at least 2 double column pages)

Exclusion criteria

1. Articles of type books, magazines, industrial reports, competitions

2. Bots designed using software engineering techniques but used in sectors besides software engineering such as Education, Business, Marketing, etc.

From the inclusion/exclusion criteria, it is clear that we wanted to filter just the articles which dealt with bots that are applied in software engineering. Since the development of bots involves software tools, the search returned several results that are not used in software engineering but used software practices. Such results do not fit clearly into the scope of the study. We removed the duplicates manually as there are two different sources of databases. Complying with the above inclusion/exclusion criteria, 70 items are filtered for further study.

Even though the search criteria are complete, to have an extensive set of research articles we applied snowballing to gather further articles. Snowballing is a method to identify additional research records based on the citations of the existing articles. Based upon the direction of tracking the citations, there are two methods, Backward snowballing and Forward snowballing. Backward snowballing tracks the mentioned references in the existing articles and forward snowballing tracks the articles where the existing articles are cited. We made use of Microsoft Academic (https://academic.microsoft.com) to obtain articles via snowballing. As the topic “Bots in software engineering” is considered, it leverages works from diverse fields such as machine learning (ML), natural language processing (NLP), human–computer interaction (HCI), etc. The citations are quite diverse and reference many papers apart from bots, whereas the articles that cite a bots in SE paper are focused on the topic. Thus, we chose to apply forward snowballing instead of backward snowballing. The snowballing technique returned 35 new records.

Figure 1 shows the screening process from initial search to final inclusion of articles for the systematic mapping study (SMS) according to PRISMA (Page et al., 2021) (Preferred Reporting Items for Systematic Reviews and Meta-Analyses)

Figure 1 PRISMA flowchart.

The study aims to answer the research question, how are the bots categorized in the wide area of software engineering. We approached the research question by answering three sub-questions: Why are bots used, what types of bots are used and how are bots designed in software engineering. These questions are answered with main classification categories such as Applications, Mode of interactions and Design principles, respectively.

• Why are bots used? Applications of bots

• What types of bots are used? The different modes of interaction with the users

• How are bots designed? Design principles used for bots

Systematic mapping process

Publication trends

The number of papers published over the years is shown in Fig. 2. The bar graph shows the papers published in the whole of computer science (CS) and in comparison, the line graph represents the papers published in the field of bots in software engineering. Since the graphs are at a different scale, the absolute values cannot be compared against each other, but the relative growth of each graph has to be noted. Evidently, bots in software engineering follows an exponential increase in contrast to a linear growth in CS. The first paper is published in the year 2016. We can see that there has been a sharp increase in interest in bots used in SE, especially over the last three years. This could in part be attributed to the BotSE workshop (http://botse.org) organised with the International Conference on Software Engineering. These points support that bots in software engineering are a recent trend, and the analysis of the research directions will provide valuable insights.

Figure 2 Number of papers published per year.

Classification scheme

We followed a classification scheme advocated by Petersen et al. (2008). Once the literature has been collected, a keywording process is used to construct a classification scheme. We went through the abstract and introduction part of all the documents one by one and collected keywords from them. These keywords are grouped in clusters and for each cluster, we labelled topics, sub-categories and categories. Figure 3 depicts the result of classification scheme with categories application, mode of interaction, design principles. These categories precisely answer the question that we are trying to answer.

Figure 3 Classification scheme of bots used in software engineering.

• Why are bots used? (Application) Software repositories, Question-Answer systems, Software development process

• What types of bots are used? (Mode of interaction) Chatbots, Collaborative bots, Voice assisted bots

• How are bots designed? (Design principles) Bot Persona, HCI concepts.

Systematic map

The systematic map reveals the frequency of publications in each categories. This encourages the readers to understand about the research activities in the past and to identify the gaps. The research items are classified based on a faceted classification, where each item may belong to one or more categories. Considering the main contribution of each article towards a category, Fig. 4 presents the proportion of papers published in each category. Almost half of the literature under study focuses on applications of bots in software engineering and the remaining half is fairly shared among design principles and modes of interaction. Overall, the highest contribution of around 27% of the published research papers are about applications of bots in software repositories. This could be attributed to the fact that the activities such as maintenance, following up on tasks and some forms of documentation can be most easily automated with the use of bots. The smallest portion of work is concerned with voice-assisted bots and bots in software development process. A reason for this could be linked to the lack of empirical studies on the need for voice bots in software engineering. Furthermore, voice-assisted bots are more complicated than text-based bots in terms of implementation, integration and usability.

Figure 4 Systematic map representing the proportion of papers published in every sector.

Every article with a major contribution in one of the main categories may also consider aspects of one or more of the other categories. For example, an article mainly describing a bot deployed in GitHub for the most part belongs to the application (software repositories) category, but the bot could also be part of a communication platform or can be used for supporting collaboration (collaborative bots). Thus, Table 1 visualises the influence of other categories in the articles towards a main category. For the articles belonging to application, around 31% of the papers also discuss Mode of interaction and the articles about design principles tend more towards applications than modes of interaction. The intersection column represents the share of articles where both the two categories are discussed. For instance, around 7% of articles with Mode of interaction as the main category discussed both applications and design principles.

Table 1 Distribution of contents from other sectors.

	Application	Mode of interaction	Design principles	Intersection	
Application	49%	31%	23%	3%	
Mode of interaction	29%	50%	29%	7%	
Design principles	50%	23%	27%	–	

Results

The categorization of bots, sub-categories and their usage in software engineering is studied in this section.

Applications

Bots in Software engineering are applied in three main areas, managing software repositories, controlling the software development process (SDP), and in question answering systems (Q&A bots). In software repositories, bots are used to manage pull requests, issues, or source code. These bots execute automated scripts when specific events are triggered. In SDP, bots help developers with recurring development tasks. These bots aim to improve the efficiency of the development process, either by improving the productivity or reducing the complexity of development workflows. In case of Q&A bots, users contact these bots to access specific knowledge bases effortlessly.

Managing software repositories

Software repositories have heavily contributed towards collaboration in software development. Repository platforms provide utilities for code documentation, discussion of issues, running continuous integration (CI) scripts and executing version control commands. Due to the standardized workflow processes of pull-based models, usage of bots is facilitated in such environments to automate recurring tasks. Bots have been increasingly adapted for managing software repositories since the year 2013. In a sample of 351 popular GitHub projects it was found that at least 13 of them make use of software bots (Wessel et al., 2018).

The general workflows of collaboration models in today’s most common versioning system (Git) is based on pull requests issued by the contributors. Every pull request contains one or more commits to the specified branch of the repository. These commits could be a fix for an issue, a feature or documentation. Some repositories are also configured with continuous integration (CI) pipelines to verify whether the received pull request passes various levels of tests so that the pull request can be merged with the specified branch. We would not consider a CI tool which merely checks for build failures to be a bot, but there exist bots which are activated when there is a build failure. Bots used in software repositories are classified by Wessel et al. (2018) into 12 categories based on the tasks they perform. These are license agreement signing, CI failure reporting, automated code review (code smell checking, test coverage, style checking), welcoming newcomers, assigning code reviewers, merging pull requests, fixing vulnerabilities, automated testing, automated software builds, controlling dependencies, issues creating, and benchmarking. Apart from these, bots can also be used for documentation, code cleanup, and code analysis. To better understand the usage of bots in software repositories, Dey, Vasilescu & Mockus (2020) studied the types of commits made by bots in GitHub. The results show that bots are used for editing documentation and configuration files much more than source code files. Specific tasks performed by software repository bots are discussed in more detail in the following.

Management of pull requests.

In a study by Wyrich et al. (2021), bots are detected from a list of GitHub repositories and the pull requests initiated by bots are compared to those sent by humans to see how maintainers view the pull requests by bots. About 13 of the pull requests were generated by bots. An interesting finding in this regard is that out of 20 million pull requests, 23 of pull requests are created by 2 million humans and 13 of pull requests are created by 5k bots. The time taken to merge a pull request when it is created by a bot is much longer (10 h) than a pull request issued by a human (14 min) and it also depends upon the author of the bot. The pull requests by bots are merged 38%, whereas human created pull requests are merged 73% of the time. Pull requests from bots have fewer comments. These points imply that either the pull requests created by bots are of low quality or these bots are not trustworthy as much as humans in the development community.

Once a pull request is sent, it goes through various processes such as code reviews and discussions. Finding an expert for reviewing the pull request is a hard task. Bots are used to find the best reviewers based on the context of a pull request. Mention-bot (Peng et al., 2018b) is an automatic reviewer recommendation (ARR) bot, which associates suitable reviewers to pull requests by tagging them. Although the bot is evaluated with a positive score, it was hard to configure, and the default setting was not great. The bot also missed providing reasons for the choice of the reviewer, which was addressed by Sankie (Kumar et al., 2019). Sankie is an AI-based service, which trains on the data from the software development life cycle to assist and provide information to software engineers. Sankie recommends reviewers in pull requests along with explanations on the frequent authorship and reviewership of the suggested reviewers.

In a study on code review bots, Wessel et al. (2020b) found that the use of such review bots is associated with an increase in the number of monthly merged pull requests and a decrease in the number of pull requests waiting to be checked. Furthermore, the time taken by maintainers to reject a pull request also got reduced, same as the communication effort around pull requests. Kinsman et al. (2021) surveyed the effects of GitHub’s features for automated workflows for repository maintainers (GitHub Actions). They found indications that after the adoption of actions, the number of rejected pull requests increased. However, these findings do not correspond well with the study of code review bots as reported by Wessel et al. (2020a). These contradicting results are due to the difference in fundamental functionality of actions and code review bots. GitHub actions include workflows such as CI, utilities, deployments, and publishing. Code review bots have the sole functionality of reviewing code. After the adoption of actions, the maintainers received faster and clearer feedback on pull requests, which led them to reject more pull requests. Whereas in case of code review bots, the contributors received constructive feedback from the bot on what has to be done to get it accepted, which resulted in increase of merged pull requests. In light of the observations described in the previous paragraph that pull requests created by bots are often rejected as they are not trusted by the maintainers, it can be said that bots are of greater utility when it comes to pull request handling rather than pull request creation.

Management of pull requests and issues is further supported concerning status tracking. In large software projects, pull requests and issues can sometimes go stale without being actively handled for a long time. In this context, Wessel et al. (2019) analysed the usage of ’stalebots’ in GitHub that automatically identify stale issues and pull requests. Among the considered projects, 87.7% adopted ’stalebot’ for both issues and pull requests, 9.8% only for issues and 2.5% only for pull requests. Even though a configuration file was provided to control the activity of the bot, 60% of the repositories added just an empty file, 6.5% maintained default settings, and for the projects with configuration file changes, 83% of them made very few changes to it. It was also found that pull requests are more susceptible to go stale than issues.

While creating pull requests, code clones are also managed with bots. CLIONE (Nakagawa, Higo & Kusumoto, 2020) is a software tool that helps to detect code clones while creating pull requests. This is an improvement to the traditional tool clone notifier, in which CLIONE detects the code fragments in improper pull requests. Improper pull requests are those in which clones were modified non-simultaneously. CLIONE is evaluated against clone notifier for the improper pull requests, and CLIONE tracked clone changes 27.5% more accurately than the traditional tool.

Dependency management tools are used to keep legacy code updated with the dependencies. Badges function as indicators to display outdated dependencies that forces maintainers to take respective actions. There are also bots in the form of automated pull requests which directly issue dependency fixes. Mirhosseini & Parnin (2017) analysed the effect of badges and automated pull requests for dependency management. Results showed that automated pull requests, despite being susceptible to notification fatigues, can encourage developers to update dependencies quicker and at a higher rate than badges. The merge rate of the pull requests increased, when the pull requests are accompanied with CI pipelines as they can reduce the chance of encountering broken builds.

Source code maintenance.

Static code analyses are run on the code to check for vulnerabilities, errors, standard violations, etc. Since they are run with the use of static analysis tools, it is further automated with the help of bots. CCBot (Carr, Logozzo & Payer, 2017) automates insertion of code contracts in C# projects. The bot involves a static code analysis tool, compiler, version control, and a CI. C-3PR (Carvalho et al., 2020) runs static code analysis on new commits and sends pull requests with explanation. The C-3PR brain module decides the corresponding tool for the newly added commits.

Source code refactoring is a tedious but needful process, thus, certain parts of refactoring processes become increasingly automated using bots. Wyrich & Bogner (2019) developed a refactoring bot, which identifies code smells via the static analysis tool SonarQube, refactors out simple warnings, and sends the refactored changes as pull requests to the developers. The bot supports limited interaction, whereby the developer can make changes to the refactorings. Serban et al. (2021) studied the different methods to propose fixes for static analysis warnings. They developed SAW-BOT (Static Analysis Warnings Bot) to provide fixes to the static analysis warnings. SAWBOT invokes SonarQube to perform static analysis and then generates a fix to those warnings and reports the suggested fixes to the developer as both via pull requests and GitHub suggestions. 4/5 participants preferred the use of GitHub suggestions, where the acceptance or rejection of the fixes can be controlled by the developer. But GitHub suggestions might not be uniformly beneficial for all kinds of software engineering bots. As the pull request doesn’t have any explanations on why such fix has been made by the bot, transparency is mentioned as a big issue. Refactoring bot (Rebai et al., 2019; Alizadeh et al., 2019b; Alizadeh, 2020) takes care of documenting the refactorings. The Refactoring bot generates natural language documentation for every pull request using template-based approaches. The bot compares the newly added commit with the previous revision and generates corresponding documentation. As the documentation is sent as PR, the maintainer could accept or reject them. The working time of the developers are also considered in this work, to ensure not to get interrupted by untimely pull requests.

Repairanator (Urli, 2018; Monperrus et al., 2019) monitors for test failures in Continuous integration (CI), reproduces them locally and then attempts to run program repair tools for a repair attempt. The fixes found from the repair attempts are sent as patches to the developers. The bot managed to reproduce only 30% of the build fails and patches are generated for 0.4% of them. Even though such patches resolved the CI fails, they were not reported to the developers, as they were results of overfitting. Also, the patches without sufficient documentation on the effect of the patches to the code are not encouraged. Further work by Monperrus (2019) addressed the problems of these dry patches. The work tackled on formulating natural language explanations for the patches along with dedicated examples regarding the behavior of the proposed patch. It also focused on communication between bots and humans, thus expecting the bots to be one among the group of developers in the future. This is fascinating because bots are not just seen as tools for automation, they are envisaged as a part of the community. Software repair bots are envisioned as a standard tool for helping developers to maintain large code bases (Urli, 2018).

Flexirepair (Koyuncu et al., 2020) is an open framework used for automated program repair (APR). The bot is built based on a software maintenance concept called generic (semantic) patch. These generic patches are unified representations of fix patterns. Armed with a template-based program repair model, Flexirepair offers a means to measure and assess new repair contributions with the proposed framework. Baudry et al. (2021) leveraged machine learning capabilities in repair bots by developing R-HERO. R-HERO uses continual learning to learn from CI builds and single line commits on how to repair the bug fixes. The model is trained with commit messages and the commit changes. Whenever there is a build failure, R-HERO checks the failed build and tries to fix them using the learned model. In performance, the bot managed to repair just 13 out of 44,002 build fails. This is because the learning model suffered from catastrophic forgetting. Even with enough data, the learning models used in the above discussed program repair bots did not learn to generalize to the new data. Often they suffer from overfitting or forgetting. Possible solutions are to explore other types of learning models by training them with abstract intermediate representations, for better generalization.

Open-source contributions from volunteers could risk a project if the commits do not ensure necessary security measures. A commit which introduces a bug in the project has to be prioritized for efficient use of the effort of reviewers. Just-in-time defect predictor predicts the riskiness of a commit. JITBOT (Khanan et al., 2020) is a GitHub bot that provides explanations for the risk factor of commits. JITBOT provides a risk metric and why it is risky. It also provides the suggested solutions for them. When a pull request is received, the JITBOT finds the commit with the highest risk factor and provides the explanations as a comment and suggestion to mitigate the risk.

Apart from analysis, refactoring and repairing of the actual source code, bots also contribute to the automation of software documentation in repositories. OpenAPIDocGen (Peng et al., 2018a) generates automatic documentation for OSS projects. This was developed using an API knowledge graph built from the source code, and the API elements are linked via background knowledge like WikiData, GitHub issues, and Stack overflow posts. In repositories, whenever the code is updated with changes, the documentation comments are left out. To update the documentation according to the code changes, upDoc (Stulova et al., 2020) has been developed. upDoc tries to fix the code-comment inconsistency by comparing the code before and after update. Using Word mover’s distance and cosine similarity, the distance between the abstract syntax tree (AST) model of the code and doc comments, the similarity is calculated.

Developer support.

Apart from content related tasks in software repositories, bots are also applied to assist novices in onboarding to new projects. To this end, Dominic et al. (2020) introduced a conversational chatbot which supports newcomers in open-source software (OSS) development in finding suitable projects and helpful resources, as well as recommendation of mentors. The bot uses Rasa for NLU, the GitHub API for content retrieval, and the StackExchange API for searching helpful community discussions.

Bot management in software repositories.

Bots in software repositories aim at increasing the efficiency of the development process by performing standard tasks or assisting developers in finding relevant information. However, bots can also create unnecessary interruptions to the developers and unwanted management overheads. Such issues were studied in detail by Wessel & Steinmacher (2020). They developed a metabot called Dashbot that effectively summarizes the outputs of several bots in a single comment. It is also a conversational bot, in the way that developers can ask questions and get responses from it. It also controls when other bots return exceptions or execute wrong actions.

Another problem that can arise by the use of bots in software repositories is that there might be a discrepancy in view on bots by software engineering researchers who develop them and developers as end users. Beschastnikh, Lungu & Zhuang (2017) addressed the issue by introducing a platform called “Mediam” to bridge the gap between researchers and developers. It is a platform for hosting analysis bots. Three vital roles are defined: Bot creator, Bot maintainer, and the developer. A researcher is the creator. They create the bot and host it in Mediam, where it is run and maintained by a bot maintainer. The developer (user) gives Mediam access to their GitHub projects, and Mediam chooses the appropriate analysis bot for the project. When the pull requests sent by the bots are merged, Mediam increases the reputation for that bot.

Q&A Bots

There is a great need to provide improved techniques for information retrieval and exploration in software engineering (Xu et al., 2017). Bots can also serve as interfaces to existing knowledge bases, aiding the developers to access them effortlessly.

Accessing information from community platforms.

One popular source of information for software developers is Stack Overflow (https://stackoverflow.com) as a community-driven forum for software related questions and answers. Around 23 of the Q&A bots studied in this section adopt Stackoverflow as the main information source. In total, around 10% of all the bots considered in this SMS make use of Stack overflow in one way or another. In this context, bot developers exploit Stack Overflow (SO) as a knowledge base and create bots that help developers to find the appropriate answers to their questions (Xu et al., 2017).

Murgia et al. (2016) implemented a Q&A bot accessing Stack Overflow to handle unanswered questions with answers from existing duplicate questions. The bot emulates a human user on the SO site. The purpose of the bot is to find a similar duplicate question for a given question, selects the one with answers, sorts them by votes and links it to the posted answer. Ilic, Licina & Savic (2020) developed a chatbot trained on SO Q&As to help users about programming questions. The bot was developed to help with Java questions, so it was trained with the 10k most popular SO questions on Java Spring tags. The bot is also integrated with messaging platforms for easier access. The bot is evaluated for the confidence scores of Intents and entities for the NLU unit, whereas the overall performance of the bot is not discussed. AnswerBot (Xu et al., 2017) also does information extraction from SO. Using a relevant question retrieval system, AnswerBot finds the relevant question from a pool of SO questions for a given developer query. The answer sections are selected based on a ranking mechanism, and the bot returns the summary of useful paragraphs. User studies are conducted to check for relevance, usefulness and diversity of the query-response pair.

As conversational systems, bots respond to developer questions from information sources. MSRBot (Abdellatif, Badran & Shihab, 2020) is an interactive conversational assistant for answering developer questions that makes use of the large variety of information that can be automatically extracted from software repositories. The system was built around a knowledge base created from a collection of frequently asked questions related to software repositories that were found in selected research papers. In the user study, 90% found it to be effective. Results also show that, the speed of the bot maybe impacted by how the user frames the question and the accuracy of NLU will improve the overall accuracy of the bot. Iterative refinement strategies are used to boil down vague questions to specific ones. Chatbot4QR (Zhang et al., 2020) is a conversational bot which provides answers to developer questions from SO through refinement. The bot helps to refine the question, so that a precise solution can be detected by the bot. The bot extracts the top-n similar questions from SO and interacts with the user by prompting clarification questions based on the tags present in the responses. The user’s feedback then adjusts the query with weights, thus resulting in a refined list of top-k similar questions. Bansal et al. (2021) presented a Neural QA model by applying deep learning methods on source code directly. Neural QA model trained with encoder–decoder Recurrent neural network (RNN) has been adopted for software engineering chatbots. The bot is trained with source code and documentation to answer subroutine questions from the developers. Apart from direct question-answers, developers are also interested in opinion Q&As. These opinion Q&As provide solutions to subjective questions from developers. To furnish such information to developers, virtual assistants need opinion Q&A datasets for the training process. ChatEO (Chatterjee, Damevski & Pollock, 2021) is a bot which identifies opinion Q&As from developer chats, emails, and issue reports. Using deep learning methods, Q&A identification is carried out, and it significantly outperformed the existing methods.

Besides being applied for information retrieval solutions, bots are also employed in learning media. TutorBot (Subramanian, Ramachandra & Dubash, 2019) helps software developers to search for technical content from Stack Overflow, GitHub, and Massive Open Online Courses (MOOCs) in a single place. The bot is built as a conversational UI (with voice support), which also attaches a recommendation engine for suggesting relevant topics as per collaborative filtering and content analysis. Evaluations revealed that TutorBot is effective, provides faster results and saves time; multi-lingual support is required for non-english learners. As the accuracy of the speech recognition systems are not so great, people with speech disorder and stammering had problems using the bot.

API discovery.

APIBot (Tian et al., 2017) helps developers to find solutions for API related questions. It is built on top of Sirius, an open end-to-end personal assistant and enhanced with domain knowledge on API documentation. The bot achieved 70.6% with Hit@5 metric in comparison to 2.2% for the baseline Sirius QA. MULAPI (Xu et al., 2019) is a tool which helps developers by recommending suitable APIs. For a received feature request, with the help of the historical related requests, the MULAPI tool recommends API methods and their usage locations. This helps the developer to know which APIs to use as well as where in the source code to use them. Developers can also interact with MULAPI for more accurate API recommendation. CROKAGE (Silva et al., 2019) (Crowd knowledge Answer generator) is a tool that provides code with explanations from Stack Overflow, given a natural language query. CROKAGE leverages the API knowledge present in SO. It uses Fasttext for fixing the lexical gap between query and SO question. Based on the API recommendation systems, API relevance scores are calculated for the answers. The candidate answers are ranked according to the weights and NLP techniques are applied for top quality answers to compose the solution.

These contributions show that bots harness the available information from sources such as SO, repositories, and API knowledge bases. Being a multifaceted discipline, the performance of the bots also depend upon other factors such as the design of NLU modules, speech recognition systems, and configuration of the knowledge base.

Software development environment

Bots deployed in developers’ local environment function as assistants supporting them in their specific tasks from modeling to testing to requirements engineering. In contrast to software repository bots that are more concerned with the collaborative nature of the software project, these bots are like personal assistants that are run on development environment and serve the developer.

Project management.

A popular bot in this strand is Devy (Bradley, Fritz & Holmes, 2018), which is a conversational developer assistant(CDA). It helps the developer to execute version control commands controlled by spoken language on the fly such that the developer does not need to switch to the Command line interface (CLI). Devy executes low-level Git command tasks based on the high-level natural language (NL) tasks from the developers. Since Devy is aware of the current context, it infers the current information automatically.

To support developers’ awareness about the current state of a project, Sharma et al. (2019b) developed the Project Insights and Visualization Toolkit (PIVoT) as a dashboard that provides actionable insights and metrics about the software project such as technical debts and assigned tickets. PIVoT serves a digital co-worker called smart advisor, a desktop buddy that contains the project’s knowledge. The dashboard runs on an XR (augmented reality) based environment that visualizes technical debt and the assigned lists as 3D metaphors. Bots used in project management do not have to function on the source-code level, but could benefit by exploiting conversations. When bots are instructed to observe specified activities, they are considered as an additional pair of eyes among the developer community. In agile development processes, Matthies, Dobrigkeit & Hesse (2019) developed chatbots that take notes from the chat discussions and points them out in the forthcoming meetings.

Testbots.

Testbots are the ones which run automated tests and static code analysis. As the bot designers and the bot users belong to separate groups, it is of importance to know beforehand the expectations of the developers. In the works done by Chukaleski & Daknache (2019) and Erlenhov et al. (2020), interviews are conducted with software developers on how to design the testbots. Guidelines are listed based on the responses from the interviews. Some of the key guidelines are, to use asynchronous programming methods to invoke system endpoints, covering test dependencies by chaining dependant tests via specific callback asynchronous functions, and to create small and modular tests.

Bots also exploited the area of domain modelling. Saini (2020) developed a domain modelling bot as a framework to teach modeling skills to novice modelers using Model-Based software engineering (MBSE). The bot extracts domain models from problem descriptions in natural language using rule-based Natural language processing (NLP), Machine learning (ML) models and generates recommendation. The bot is not a conversational bot, rather a Graphical user interface. The extracted domain models are not accurate enough to be used directly in software development. Thus, Saini et al. (2020) extended the work by developing DoMoBOT (Domain Modelling Bot). DoMoBOT is interactive, and it allows the developer to update the domain models once they have been created. The modeler chats with the bot to obtain insights into the modeling decisions taken by the bot.

In software development, requirements specify software behavior and tests validate the fulfillment of these behaviors. Specbot (Pena, Cabot & Gomez, 2018) automates the translation of requirements into tests code. The requirements are written in a structured fashion using Gherkin (https://github.com/cucumber/cucumber) notation, the NLP unit matches the requirements to intents and entities, which are then sent to Bot logic unit, where they are matched with executable specifications, written in RSpec (Chelimsky et al., 2010). The context management unit takes care of the conversation based on previous utterances.

Mode of interaction

Based on the interactive nature of the bots, they are classified as chatbots, voice bots and collaborative bots. Chatbots are purely text-based bots. They respond to a textual input in the form of text output, or sometimes in graphical content. They are used either in chit-chat for entertainment or to execute a specified task. Voice bots are ones which let you access task-oriented bots via a speech interface. Another mode of interaction is the bots used for collaboration. These bots are developed to work among several developers and not for one-to-one conversation. The bots are deployed in communication channels and are predominantly used by developers for discussion and collaboration.

Collaborative bots

Collaborative bots are conversational chatbots which run on communication/chat platforms such as Telegram, Slack, Twitter, or Discord. As the name suggests, these bots foster collaboration among the developers. Apart from the bots which converse with a single user, these bots are aimed to be a part of a discussion among a community. Being reactive, they retrieve useful information during the discussion, whereas proactive bots provide interesting tips based on the topic of discussion. As most of the collaborative bots are reactive, they do not interfere in the discussion among the peers.

SOCIO (PerezSoler et al., 2017) is a conversational modelling bot used for collaborative modelling among various levels of developers. SOCIO helps to create and design software models through Natural language commands. It is deployed on Twitter and Telegram channels. This facilitates bot developers and designers to model collaboratively through social media channels. They improved the bot further (PerezSoler, Guerra & de Lara, 2018) by considering the decision-making processes for multiple possibilities to reach a soft consensus among everyone. In the study by Ren et al. (2020), SOCIO has been evaluated against an online GUI-based tool and SOCIO performed great in terms of efficiency and satisfaction, and equivalent in effectiveness and quality. In contrast to traditional modelling, PerezSoler, Guerra & De Lara (2017) and PérezSoler et al. (2019) proposed social conversation with model development through chatbots and conversation as a platform (CAAP). Feasibility of the work is demonstrated with a case study that focused on collaboration in natural language, multi-platform, mobility, and traceability.

Taskbot (Toxtli, Monroy Hernández & Cranshaw, 2018) is a communication bot deployed in Microsoft Teams, where users can tag the bot and create tasks which will be communicated with other users and keeps remainder. With an overall positive score, some negative feedback included are unable to recognize people names without proper syntax, highly frequent reminders, handling multiple threaded conversations, and human ambiguity. Cerezo et al. (2019) developed a bot which recommends experts in a software community in a Discord chat. The results unfortunately turned out negative, as the bot did not consider the social aspect of carrying out a conversation. As no help instructions were provided, the bot was found non-user-friendly. Bots also function as an interface to access software visualizations. Bieliauskas & Schreiber (2017) developed Sofia and deployed it in a communication platform for developers to interact with visualizations. The developer can directly ask Sofia about a ticket, and the bot actively serves a customized visualization based upon the request. Being on a communication platform, Sofia also monitors passively when two or more developers are discussing a subject and provides a suitable visualization for them.

A significant feature in collaborative bots is that they can passively listen to the conversations and perform actions according to their functionality. GitterAns (Romero, Parra & Haiduc, 2020) is a bot deployed on Gitter (https://gitter.io) channel that provides solutions to the problems faced by developers. The bot passively listens to their conversations and when troubleshooting problems are discussed in the chat platform, GitterAns identifies which turn of the conversation is related to the problem and provides a solution appropriately. To enhance the productivity in a collaborative working environment (CWE), Corti et al. (2019) adopted a virtual assistant to assist developers by leveraging Internet of Things (IOT) and instant chats. The bot listens to the chat messages of the developers, interprets them and executes the respective actions. The core aspects of the assistant are collaboration, communication, information management and content over structure. Most collaborative bots are reactive, but being reactive in a communication channel might not mean that the bot has to respond to the main line of discussion. There are also bots, which attend to specific keywords in the overall topic of discussion. Sannikova (2018) deployed a chatbot on Slack, and it is connected to a restaurant menu page. Even though the main topic of discussion will be software development, the bot only responds to the topic of food. The bot returns the day’s menu, When the developers query about food. Being a reactive bot, it will not interfere with other discussion topics.

During collaboration on closely related tasks, it is often hard not to have conflicts with the code developed by others in the team. Code conflicts are usually managed with best practices, but using bots for eyeballing the conflicts eases up the work of developers. Sayme (Paikari et al., 2019) takes care of code conflicts by notifying the developers proactively and reactively. It constantly looks for conflicting files and notifies developers on such conflicts. One can also ask Sayme about the status, and it provides results reactively. The most notable thing about Sayme is that it detects code conflict even when the conflicts are in different files but shares common code execution like function call and definition.

In summary, collaborative bots combine conversations and commands, thereby making bots inclusive of the development community. They are sensitive to their social aspects as they are established among a community of developers. Bots with reactive characteristics only function when the developer demands it. This serves the benefit of avoiding unnecessary notifications and noise.

Chatbots

Bots involving natural language conversation falls under Chatbots, previously called Chatterbots. Chatbots are recently being employed in various domains from restaurants to e-commerce to customer helpdesks. In this section, we discuss chatbots that are being used in the software engineering domain. Based upon how they converse, chatbots are either rule-based or generative. As the name signifies, rule-based are formed of rules or intents. The number of intents is fixed, and the user’s input could be only classified into one of these intents. On the contrary, generative models do not rely on fixed set of intents but are learned via machine learning or deep learning (Sutskever, Vinyals & Le, 2014; Cho et al., 2014). And the conversation responses are generated based on the model weights and input sentences. Generative models are still in research phase, as the responses are unpredictable, whereas rule-based models even with limited functionalities are found in most of the current products due to its reliability.

Rule-based or Retrieval-based chatbots consists of three main components: Natural language understanding (NLU), Dialogue management, Natural language generation (NLG). NLU may sound sophisticated, but essentially NLU classifies the input to one of the many listed intents and extracts entities. Dialogue management takes care of the path of the conversation. NLG generates natural language responses through templates. When the chatbots are expected to just chit-chat, no knowledge bases are needed as the commonly used chit-chat combinations are written in the Dialogue management itself. Some popularly used NLU components are Google Dialogflow (https://cloud.google.com/dialogflow), Microsoft LUIS (https://www.luis.ai/), IBM Watson (https://www.ibm.com/watson), Rasa (https://rasa.com), Xatkit (https://xatkit.com/). These components provide both NLU and dialogue management services. Abdellatif et al. (2021) evaluated the NLU components that are suitable for software engineering tasks. The comparison resulted in IBM Watson being ranked as the best for intent classification and entity extraction, whereas Rasa ranked the best for confidence scores.

To study about the activity of Chatbot topics in SO community, Abdellatif et al. (2020) mined SO for questions and answers related to chatbot tags to answer three questions, topics in the area, types of questions and difficult questions. The common topics were integration, development, NLU, user interaction, and user input. The difficult to answer questions were related to training the models. Training NLU model involves suitable datasets, frameworks, type of model. In the view of developing virtual agents (VA) in SE, the NLUs have to be trained with intents and examples from software engineering field. Since the SE-dataset is domain-based and scarce, the work by Wood, Eberhart & McMillan (2020) focused towards applying transfer learning on a global encoder –local encoder model to the task of classifying dialogue act types. The global encoder is trained on a generic dataset such as AMI business corpus and the local encoder is trained on a manually annotated small software engineering dataset.

To develop bots in SE, it is essential to design the bots to suit the needs of the developers. Wizard of Oz (WOZ) experiments are conducted to understand the expectations of the developers and design bots accordingly. One such WOZ experiment is conducted by Wood et al. (2018) to study about different speech act types used by software developers on bug repair. A supervised model is trained to classify them. Among all the speech act types, developers asked clarification questions the most. Another WOZ experiment by Melo et al. (2020) studied how chatbots will aid software developers in day to day tasks. The results demonstrate that the use of chatbots will be certainly helpful in software engineering and the role of chatbot as a guide is preferred over executing tasks for the developers. Participants also liked the contextual information that the bots possess on what the developer is currently working on.

Chatbots perform as conversational interfaces to existing databases, thereby allowing the user to access the information handily. The bots are linked to such databases and NLUs are designed accordingly. Matera & Castaldo (2019), used chatbots for data exploration, where the data is stored in a relational database. The bot replaces the SQL command line with a conversational interface. By the use of conversational relationships, qualifiers and values, the bot functions with rule-based system but with more relaxations and flexibility to the database structure. Chatbots engage developers by being an interactive interface to static documentation pages. OpenAPIBot (Ed-Douibi, Daniel & Cabot, 2020) is a chatbot that helps to provide information about the desired API. The bot offers interactive capabilities to the API navigation in contrast to the simple descriptive API documentation. The bot helps developers to understand REST APIs by asking questions using natural language. The implementation uses XATKIT and powered by Dialogflow for the NLU part. In software crowdsourcing tasks, bots apply information retrieval to help users find appropriate developers for the task. CrowDevBot (Ni et al., 2019) is a conversational bot used for Crowdsourcing. The bot assists users in completing crowdsourcing tasks via conversation, rather than tedious mouse clicks. To obtain domain information, the chatbot is linked to a database called Software service knowledge base (SSKB), which contains information about projects, required skills and experts.

Chatbots are applied in educational platforms to teach students certain skills. Boubekeur & Mussbacher (2020) designed an interactive domain modeling assistant that assists students with Domain modeling through gamification and live feedback. The functionality of the bot is described with three modeling scenarios, mistakes made with software engineering terms, placing attributes in wrong class and mistakes in player-role patterns. Chatbots are used to facilitate non-technical users as well to use model-driven engineering. iContractBot (Qasse, Mishra & Hamdaqa, 2021) is a chatbot developed with XATKIT framework for developing smart contracts. Based on Model driven development, the bot converses with the developer and develops model from Natural language interactions. PérezSoler et al. (2020) synthesized a chatbot using xatkit framework for model query. With the help of domain-meta model and natural language configuration, the xatkit generates an executable chatbot, which is then used to query models. The actions are written for every intent as execution rules.

Chatbots that help developers are not restricted to code related tasks, they also take part in productivity-based works. Amber (Kimani et al., 2019) is a proactive chatbot, that takes care of task scheduling, task reminders, managing time spent outside of work in social media platforms. It helps to review the scheduled tasks by the end of the day. It uses Microsoft bot framework and conversational learner framework. Though the bot received average ratings, the participants made changes to their routine to be more productive and healthy.

The fact that chatbots being used as an interface for accessing database in many works show that natural language interactions are actively researched over conventional methods in information retrieval. The availability of various NLU and dialogue management frameworks surely contributed to the studies in this topic. Even with the restricted functionalities of retrieval-based NLU techniques, the results are positive overall. Thus, improvement in NLP representations and ML models with resulting advancements in NLU will push the boundaries of chatbots further.

Voice assisted bots

Voice assisted bots are activated using natural language voice commands from the developer. Automatic speech recognition systems (ASR) are integrated with the existing chatbot frameworks to extrapolate the usages to spoken language capabilities. As bots are aimed to work along with the developers in a bot-ecosystem, these bots accelerate the development in such directions by leveraging voice capabilities.

Devy (Bradley, Fritz & Holmes, 2018) is a conversational developer assistant, which executes version control commands. It works as a replacement to the traditional command line interfaces. Devy is built using Amazon Alexa framework, and it is deployed in Amazon Echo device. Robin (Da Silva et al., 2020) is also a voice controlled bot for developers to access repository-based information. Compared to Devy, Robin focuses on questions useful for developers during collaboration. Robin has been implemented with Alexa and Google Assistant. Smartadvisor (Sharma et al., 2019a) is a conversational interface(supports both text and speech inputs) to aid developers with information. By connecting with a data exhaust like IDE, version control and CI, the bot could provide information to the developer via a guided conversation.

Speech interfaces are not just used for a direct transcription to the existing commands, rather with the power of NLU, they could be directed to initiate actions. This opens up many opportunities in Human–computer Interaction (HCI). IslandViz (Misiak et al., 2018; Seipel et al., 2019) lets developers visualize complex software architectures in 3D via Augmented Reality (AR). Speech-based commands are used to navigate and explore specific parts of the visualization in Microsoft Hololens. With Speech and gesture control inputs from the user and the corresponding visualization changes in the View, it provides an engaging interface to the users for exploration. Rasa is used as the chatbot framework and Microsoft in-built ASR is used for Speech-to-text (STT). Visualization of software analytics using city metaphor in AR is also done by Baum et al. (2020). The users interact with the visualization via gesture and voice controls. Although the voice control worked well from a technical point of view, it was rejected by the users and the virtual on-screen keyboard was also unsuitable.

Apart from exploring code files, code is also written using utterances. Convo (Van Brummelen et al., 2020) is a voice-based conversational programming platform. Developers issue natural language commands via speech utterances, and they are converted to respective code. Convo contains voice-user-interface (VUI), NLU module, Dialogue Manager and a Program editor (PE). The number one recommendation of the work was to improve the speech recognition system, as the accuracy of the ASR directly affects the conversational programming. Voice-Driven Modeling (VDM) has been envisioned by Black, Rapos & Stephan (2019) to adapt voice-driven programming to Software modeling. The VDM framework consists of ASR, NLP and context-specific modeling blocks. Being a position paper, the framework is aimed to be useful for software modelers to improve their modeling efficiency, to domain-experts with limited modeling knowledge, and to modelers with disabilities.

Comparatively, few works have been done on bots with voice assistance. These bots depend upon the performance of ASR systems, and speech recognition is a huge research topic of its own. These recent works on such a multidisciplinary topic will open up new research ideas and possibilities towards the use of voice activated conversational AI in software engineering.

Design principles

Design aspects of bots are grouped under Bot persona and Human computer interaction (HCI) concepts. These features focus on the social characteristics of the bots and how they are being perceived by the developers. Bots evolved from being a simple tool which helps the developers, to being almost one of them. Bots manage developer activities such as conversing with other developers, making changes, editing code and writing documentation. When Bots play such an integral part in the development process, it is significant to consider the social aspects of the bot for it truly to be accepted among the developers.

HCI concepts

To function optimally, designing a bot is as significant as developing the bot. As they function as a medium between developers and software programs, the design concepts used in human–computer interaction (HCI) takes prominence. Imparting suitable design principles for the bot and in the environment helps the bot to achieve efficiency for the intended task.

The socio-technical aspects of bots are addressed in Lebeuf, Storey & Zagalsky (2017). The work lists all the social frictions that might occur in Teams interactions, Teams interaction with Technology and individual interaction with technology. The work also listed all the existing bots which tried to reduce these frictions. Pinheiro et al. (2019) surveyed developers on the motivation and challenges using bots. Around 65% of the participants worked alone while developing the bots. The survey indicates that, most of the developers created bots for their personal needs and to assist them in the workplace. There were considerable difficulties in bot implementation and finding tools for bot development as well. Washizaki (2020) presented a vision called value co-creation of Software by AI and developers. When applying AI to development process, aspects such as automation, accuracy, robustness and explainability have to be considered. Cooperative activities and interactions are necessary among developers, among AI, and among AI and developers towards value co-creation.

Digital nudging is a concept from behavioral science, where nudges and notifications guide developers towards best practices. Brown (2019a) and Brown (2019b) applied these digital nudges in tool recommendation to suggest static code analysis tools for Java projects. By integrating Nudge theory in a bot, Brown and Parnin developed nudge-bot (Brown & Parnin, 2019), which provides recommendations about better software practices to university students via digital nudges in both active and passive fashion. Further work (Brown & Parnin, 2020) focused on developer recommendation choice architecture of the bot such as actionability, feedback, and locality of the nudges. Results from the experiment show that, developers prefer actionable recommendations to non-actionable pull request comments.

Wessel (2020) proposed the concept of a meta-bot which manages all other GitHub bots. The author suggests a study to understand how developers view bots, and provides guidelines to create the metabot. Using Design fiction as the concept, the study has been conducted among software developers to see if the vision of the meta-bot suits the need of the developers. Wessel et al. (2020b) carried out another study among OSS maintainers to check the comfortableness while using code review bots. Results show that the main reason to use such bots are better feedback to developers and reducing maintainer’s effort. Bots reduce the manual labor from maintainers to maintain a project. They also report on improvement in code quality, increase in contributor’s confidence, decreased time to close a pull request. Liu, Smith & Veeramachaneni (2020) studied the behavior of Bots in OSS projects and how it affects the interaction between bots and humans. Design principles are proposed for a trouble-free human-bot interaction. They are, to be robust and stable, ensure transparency, provide simple responses, reduced interruptions, rich user interface, personalized behavior and not to overuse bots. These principles should be used as guidelines to evaluate the user experience of a bot’s interaction behavior. Consolidating the design of repair bots, Van Tonder & Le Goues (2019) proposed 6 principles on how repair bots should be built. The principles are based on syntax generality, semantic validation, testings and static code analysis. The work also pointed out the importance of integration of human workflows in the automated system.

For a generic NLU model on a retrieval-based chatbot, bots execute the same intent every time when different expressions of the same query is posted. Users get annoyed of this behavior, as bots are insensitive to the expertise of the users. Performobot (Beck et al., 2020; Okanović et al., 2020), which runs load testing in Software systems tackled this issue by having intents at various levels. The bot lets expert, novice and intermediate users to execute the same action via different intents. This design feature gives the flexibility to access the bot according to the level of expertise.

Wessel et al. (2021) did a study on bot developers and maintainers to check the disruption of the bots to the community. Noise is found as the recurrent and central challenge, which affects both human communication and development workflow. Noise is viewed as a bot behavior, such as the number of bots in a project (overpopulation), information overload of the bots –which interferes with the collaboration model of humans on social coding platforms. When contributions made by bots are without proper explanations, transparency is an issue on why such fix has been made by the bot (Serban et al., 2021).

When developing bots for collaboration, it is important to consider the communication style of the developers. Taskbot (Toxtli, MonroyHernández & Cranshaw, 2018) had issues with identifying people names without proper syntaxes, human ambiguity and handling multiple threaded conversations with their corresponding tasks. SOCIO (PerezSoler, Guerra & de Lara, 2018) is used in a collaborative environment for modelling, where achieving a unified consensual decision among different expert groups is hard. SOCIO ensures reaching a soft consensus between modelling experts and domain experts on the task of domain-specific language modelling. Saadat, Colmenares & Sukthankar (2021) developed a bot classifier to identify bots in GitHub. They then conducted a study among humans-bot teams and humans only teams to examine the nature of workflow. The human-bots team had an active communication via issue comments throughout the development process, compared to the humans only teams. “Bots force human members of the team to discuss issues between different stages of the workflow”

HCI concepts from cognitive science such as behavioral sciences are used in the design of bots. Studies on behavior of bots provide insights into how they are perceived by the developers and maintainers. Based upon the application, appropriate design features have to be considered, as discussed in the collaborative environment section. Noise (too many notifications and untimely interruptions) and Transparency (explainability of the actions) are recognized as notable issues.

Bot persona

Bots emerged as a way to automate scripts, but with successful integration of Natural language tools they are now used to execute tasks on the behalf of humans via commands and conversations. For such interactions, it is essential for the bots to possess a persona. Persona is the characteristics given to the bot for a smoother social interaction. This comforts and even excites the humans to have a conversation with a bot. These personas are designed according to the task at hand so that it emulates the behavior of a person in that place. The bot’s persona has to be stable and consistent throughout the entire conversation.

Bots are expected to emulate social roles like humans, when bots are to be integrated into communities. This motivated Seering et al. (2019) to study about multiparty bots. Various bot personas such as antagonist, archivist, authority figure, dependent, clown, social organizer, storyteller are discussed. Kuttal et al. (2020) designed a conversational agent (CA) for pair-programming task through WOZ study. Results convey that the CAs are to be developed with social intelligence, to include embodiments (face and voice) and trustable traits. Moreover, when Agents interrupt humans, humans don’t like it unless there is a high-error probability in their computation. Developers have a welcoming and forgiving attitude towards software bots when they have a positive name and character (Monperrus et al., 2019).

Designing the persona did not purely rely on the sociological aspect of the bot. When the personas are linked with the capabilities of the bot, the persona could also reflect the skill of the bot. Erlenhov, Neto & Leitner (2020) classified Devbots based on the personas as chatbot persona (charlie), autonomous bot (alex), smart persona (sam). Currently, lots of bots with charlie and alex personas are present but very few sam type bots. All classes of bots are in some way prone to producing noise. Paikari & Van der Hoek (2018) compared the chatbots for developers based on type, guidance, direction, interaction style and communication channel. The Task-oriented chatbots don’t engage in random chatter to build relationships, as they are mainly built for executing specified tasks. None of the chatbots with behavior evolution are designed specifically to support software development. Even in collaboration, chatbots did not exhibit an engaged interaction style, but just informational and automation chatbots existed. But depending upon personal preference, the user expectations may vary. In the work done by Matera & Castaldo (2019), even the chatbot which is built for data exploration received feedback to have chit-chat nature for a better persona.

Bots sometime take the role of a developer and try to act as one among the other human developers. This leads to the discussions in the ethical point of view. There are valid arguments on both sides whether bots should disclose the identity or should it be disguised as a developer. Being in the disguise, bots have the trust of the fellow workers and are considered as one among them. This might raise questions on accountability. As being explicitly disclosed as bots, they are not taken seriously by the developers due to the limitations of the bots. Murgia et al. (2016) conducted a study by letting a Q&A bot run in the SO site for 90 days. In the first phase it was disguised as a human, and it performed well. But in the second phase where it was explicitly mentioned as a bot, the bot received a lot of downvotes, poor comments and got banned in 25 days. From the qualitative study by Wyrich et al. (2020) on a refactoring bot, the conclusion was that bots are accepted as bots, even though they are considered critical compared to human developers. This concurred with the Dagstuhl seminar discussion (Storey et al., 2020) “The bot should not hide its identity from the users”.

Apart from the ethical viewpoints of the developers, how bots should be disguised, it is vital to the user community to recognize whether the user is a bot or not. This ensures better transparency in the bot-ecosystem. Based upon the actions executed by the identity, it could be identified as bot or not. To identify bots, Dey et al. (2020) proposed BIMAN (Bot identification by commit message, commit association and author’s name). Based upon the characteristics of the commits, they are scanned for unusual behaviour which are abnormal for human developers. They also considered the time and spike of the bot’s activity for better identification. Golzadeh et al. (2020) did a similar work on identifying bots from human contributors based on the comment patterns in pull requests. Based on a combination of Levenshtein distance and Jaccard similarity of the comments, they are clustered and bots are identified. It is based on the hypothesis that bots repeatedly use a certain set of words. Further work on the bot identification by Golzadeh et al. (2021a) used encoded representations with ML methods for classification. Based on the structure of comments from pull request and issues the bots are identified by training an ML model of vectorized words with Bag-of-words (BOW) and Term frequency Inverse document frequency (tf-idf) techniques. Golzadeh et al. also published a ground-truth dataset (Golzadeh et al., 2021b) of 5k GitHub accounts with bots tag to promote research and evaluation of bot identification methods.

In summary, the persona of the bot has to be designed in accordance with the role and functionality of the bot. Oftentimes, it is also restricted with the capabilities that the bot possess. Social aspects of conversation are expected even from the task-oriented bots. Should bots disclose their identity or is it ethical to be in disguise of a human, still remains as an interesting discussion. Nevertheless, the tools, which are used to identify bots are actively developed.

Discussion

This article presented a consolidated study of bots used in software engineering. It is clear from the above sections that active research is happening in this field. From running on local machines to being deployed in cloud or server environments, bots have secured a place in modern software development processes. In applications point of view, bots are employed in managing software repositories, aiding software development process and accessing information in Q&A systems. Based on our survey it appears that bots are most established in the area of software repository management. This is not surprising, given that software repository management becomes increasingly automatised trough scripts and well-defined workflows that can be executed by bots. However, with the recent advances in artificial intelligence in the area of knowledge acquisition and management, it can be expected that in the future bots will be more utilised as digital companions that support developers in ad-hoc finding, accessing, and structuring information in technical documentations and in community discussions.

The modes of interaction discussed the ways in which bots are controlled, such as conversational chatbots, collaboration bots and voice activated bots. Developers used chatbots as natural conversational interfaces to retrieve information, which enhanced the overall user experience in accomplishing a task. Bots used in collaborative environments are designed to execute tasks among the developer community. To consider a software program as a part of a community is a great accomplishment and privilege. This level of cooperation demands the bot developers and designers to consider different sociological aspects. Trust becomes a major factor here, as the decisions taken by the bots have to be trusted by the fellow members. Bots could handle this issue by furnishing sufficient explanations for the decisions taken. Whether bots affect the workflow of software developers positively or negatively depends on the capability of the bot. For a smooth collaborative scenario, bots are expected to be smart. Bots need to understand the context and provide a satisfying experience to developers (Wyrich et al., 2020). Even the contributors in open source software repositories expect the bots to be smart (Wessel et al., 2018). For the bots to achieve such smartness, developments in frameworks, algorithms, NLU, NLG are required. In the current state, people do not seem to trust the solutions from bots in any case, perceiving humans to be superior to bots in computer-mediated communication (Murgia et al., 2016). Smartness for bots is a double edged sword, that it has conflicts with trustability. As bots get smarter, it is equally important to make the models explainable. Bots with autonomous and smart persona are hard to trust compared to chatbot-style bots (Erlenhov, Neto & Leitner, 2020).

With the success of voice controlled digital assistants in households, research has also explored how software engineering bots can be equipped with automatic speech recognition such that developers can interact with them in a natural fashion, during their works. One major challenge in this regard is to adapt automatic speech recognition models that accurately understand the very specific language used by developers. Without sufficient accuracy of such models, using voice controlled bots can be frustrating, especially for those with non-native accents when commands have to be repeated and cognitive load increases (Van Brummelen et al., 2020). Studies (Winkler et al., 2020) show that rather than being entirely dependent on the voice input, a combination of voice and text based system overcomes these challenges. With dynamic research of DL methods and language models in speech recognition systems, it is assured in the near future to have increased accuracy and domain adapted speech models. This gives rise to digital companions, to which developers can converse about the tasks on their day-to-day basis.

The design principles of the bot take into account the persona that is given to a bot which contributes to the objective one tries to achieve with the use of the bots. In most of the implementations, the bot development process is given attention but not in designing a persona. Eventhough bots are regarded as one among the community, they should be identified as such and not in the disguise of humans. As in these initial stages of research, it is reasonable that explicit mention of bots might influence the participants to have biased opinions. But with sufficient awareness of the capabilities and decision making process of the bot, this bias could be eradicated. One of the main reason why bots are used in software engineering field is that, in software engineering there are monotonous tasks which are mundane for developers to execute them. As technology progressed, such undemanding tasks are left to scripts and algorithms to be automated. Consequently, the reasons why people use bots are: Problem notifications, getting rid of undemanding but disturbing tasks, and information integration (Erlenhov, Neto & Leitner, 2020). Negative effects can occur as well when too many tasks get out of the hand of the developers. For example, users who tried out Taskbot (Toxtli, MonroyHernández & Cranshaw, 2018) felt that the remainders from the bot were annoying and too frequent (both spatial and temporal). These are considered as noise, as they do not contribute effectively to the development. These factors enforce the bot developers to consider the social aspects of the bots while designing them. One could design them reactively with limited autonomy, as it would not pose any problems related to flooding of notifications. However, in the work by Cerezo et al. (2019), the feedback was that Bots should be proactive rather than being reactive. This suggests, that much more research in the design of software engineering bots and how they effectively collaborate with developers is needed.

Considering the current available bots in software engineering, they are often not that smart and cannot execute all the queries that the users wish. Hence, for the developers interacting with bots, it is essential to know the capabilities of the bot, which commands and actions are possible and how they could be triggered. Feedback from the data exploration chatbot (Matera & Castaldo, 2019) mentioned that the bot should provide hints or help at first and it has to support complex queries. Users who used Performobot (Okanović et al., 2020) also mentioned that the bot should provide information about available commands, hints and available keywords. The bot lacks explanations and is unsuitable for complex load performance analysis for experts. Supporting complex queries might have to do with better NLU and sophisticated frameworks, but exhibiting the features and limitations to the user is certainly an obtainable task. Being an interdisciplinary topic, the performance of the bots depend upon other intertwined research areas such as sociology, human–computer interaction, natural language understanding, speech recognition systems and artificial intelligence. As there are exploration and analysis in all these fields, it will only accelerate the progress of bots in software engineering.

Research gaps

Research on bots in software engineering began by automating tasks through scripts mainly on software development level, but with time the field has become much more interdisciplinary. A fully-fledged development process consists of designing the bot (persona, behavior, mode of interaction), and implementing their abilities (training of machine learning models, development and deployment). The research that has happened in these sectors were discussed above. However, it became evident that there are still some aspects that requests further research.

Here, we list some of these research gaps that we identified from writing this paper along with potential research directions and challenges that follow them.

1. This study turned out that the design of interfaces and interaction design plays an important role for the acceptance of bots in software engineering with respect to transparency and controllability of bot’s actions. While most of the literature reports on the technical development of bots, this calls for research from the field of human factors and cognitive science to develop the next generation of software engineering bots that are perceived as added values and smoothly integrate into the workflows of software developers. Even bots that work autonomously need to be equipped with interfaces to be observable and predictable by users. Current interactive bots use mostly textual chat interfaces. As already mentioned above, speech controlled interface is a potential domain to be explored. With rapid progress in speech recognition systems and spoken language understanding, bots which are limited with textual inputs could be supplemented with recognizing spoken utterances so that they do not necessarily have to be controlled via key-press and mouse inputs on desktop screens. This would allow users to speak to a bot while doing something else, like coding, without the need to switch from the development environment to another system. From an HCI perspective this would also constitute a positive step in terms of social aspects of bots as digital companions. It is a great sign that there is live research in designing the persona of the bot. Since, bots are supposed to work alongside human developers it cannot be regarded anymore just as a tool. To ensure non-erratic and pleasant behavior, bots are modelled with personas. Also bots have to consider the social conditions of fellow developers when executing any actions. Armed with speech controls, bots could even be run on virtual and augmented reality screens.

2. From implementation perspective, there is still a problem of adopting automatic speech recognition for the software engineering domain. Generic ASR models trained for day-to-day use cases often fail in recognising specific language used in the software engineering domain. Since well curated audio datasets for the software engineering are difficult to acquire, one alternative is to use existing speech recognition models and fine-tune them to be able to understand the language of software developers.

3. Current chatbot frameworks for dialogue management focus mostly on the rule-based models as opposed to learning based ones. This is mainly due to the feasibility and precision of the bots. In the current stages, the intents and entities classification are already powered by ML models. With the expectations of smart bots in every sector including software engineering, it is valuable to leverage the flexibility of machine learning based models in addition to the reliability of retrieval-based models in dialogue management and natural language generation.

4. Bots getting autonomous and smarter will be a reflection of the research and improvements in Natural language models, Machine learning and AI, even Artificial general intelligence (AGI). However, as there is progress in technology there is a significant amount of fear related to safety, responsibility and trust. These concerns are handled in the context of explainable AI, contributing towards transparency, verification and robustness. This is currently a necessary field of research for steering the future of bots in the ethical course. Considering the characteristics of the bot personas, empirical studies in the intersection of sociology, psychology and computer science are needed to understand the expectations of the user community about bots.

Conclusions

The systematic mapping study laid out the answers to what, why and how are bots used in software engineering. In the explored areas, lots of active research happened in managing software repositories and insufficient works have been done in voice controlled bots. Design aspects are important part of bot development and are often disregarded. These insights will help the researchers and developers to understand the current status of bots and encourage to push the boundaries of bots further.

Future work

• This work always can be improved with the addition of new articles and the bots can be categorized based on other dimensions.

• Grey literature review could also be tried out, since there are lots of interesting articles outside of academia.

• Review of bots in practitioner’s view through empirical studies and other domain such as sociology and human factors could offer a new perspective to the current work.

Supplemental Information

Supplemental Information 1 Charts representing the proportion of articles published in each sector

This data was used to generate the charts.

Click here for additional data file.

Supplemental Information 2 The python file used the data source file to generate the bar and line plot of papers published over the years

Click here for additional data file.

Supplemental Information 3 Data source

This dataset was obtained from dblp: Publications per year (https://dblp.org/statistics/publicationsperyear.html).

Click here for additional data file.

Additional Information and Declarations

Competing Interests

Author Contributions

Data Availability

Stefan Wagner is an Academic Editor for PeerJ.

Sivasurya Santhanam conceived and designed the experiments, performed the experiments, analyzed the data, performed the computation work, prepared figures and/or tables, and approved the final draft.

Tobias Hecking analyzed the data, performed the computation work, authored or reviewed drafts of the paper, and approved the final draft.

Andreas Schreiber analyzed the data, authored or reviewed drafts of the paper, and approved the final draft.

Stefan Wagner conceived and designed the experiments, performed the experiments, authored or reviewed drafts of the paper, and approved the final draft.

The following information was supplied regarding data availability:

The classification diagrams and the plots are available in the Supplemental Files.

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
