# Peer review of "Bots in software engineering: a systematic mapping study"

_PeerJ Computer Science, doi:10.7717/peerj-cs.866_

## Round 0.1 · original submission · Minor Revisions

Good work, but you need to address the reviewers' suggestions, particularly those concerning technical details and existing approaches. Please revise the article and re submit.

Reviewer 1 ·

Basic reporting

The introduction provides a good, generalized background of the topic Bots in Software Engineering that gives an appreciation of the wide range of applications for this technology. However, the motivation for this study need to be made clearer.

Experimental design

The survey methodology is consistent with a comprehensive coverage of the subject.

Validity of the findings

It may be worthwhile to mention the tradeoffs involved in choosing the inclusion/exclusion criteria as opposed to other criteria to filter out irrelevant articles to the mapping study

Additional comments

Overall, this is a clear study manuscript that has implications for the development of Bots in Software Engineering

Reviewer 2 ·

Basic reporting

no comment

Experimental design

no comment

Validity of the findings

no comment

Additional comments

In this Paper complete information related to bots with empirical data provided.
it is a good source of information for developing new bots.

·

Basic reporting

In general, the article is written in a clear and reproducible manner. There is an overall storyline, which supports the reader. Besides few minor issues, however, the report lacks information in two key aspects.

Minor issues:
- There is an inconsistency: the cover page abstract talks about a systematic literature review, while the rest of the paper talks about a systematic mapping study (more on that in the research design section).
- Since this article does not have any page restrictions, authors should avoid the heavy use of abbreviations, e.g., SE, NL PR etc. Furthermore, it happens fairly often that, when abbreviations are used, no spaces have been left. For example, p. 2, l. 55: “study(SMS)”. Authors should thus carefully read the text and fix those issues.
- In section 1 (introduction), authors should provide a bit more structure. A structured introduction with subsections “objective”, “problem statement”, “context”, “contribution” would help better support the reader getting key information quickly.
- Also, in section 1, authors should provide a subsection “outline” in which they guide the reader in terms of what can be found where in the article.
- Fig. 2 is of limited use as the authors do not provide numbers at the bars. Hence, this whole figure can be used as a trend, yet, authors should add the numbers to the bars and also to the trendline in order to provide details required to understand/follow the text.
- Fig. 3 and Fig. 4: Authors should revise these figures. In their current form, especially with the color scheme applied, the figures are (in the best case) hard to read (in a gray-scaled printout they are unreadable). If I may, I’d suggest converting Fig. 3 into a tree to better visualize the topics and hierarchies. Fig. 4 should be converted into simple tables.

Critical key aspects: Even though the general reporting is of good quality, the article suffers in two key aspects:
1) Related work: the related work should be revised substantially. About a third of the related work is only concerned with pointing to an ICSE workshop and dropping the 3 categories discussed on this workshop. It continues with collecting further terms, but the related work does not provide a detailed description. I wonder if something like Fig. 5 would be beneficial here. In particular, I miss a clear approach in setting the scene in the related work. Few aspects are mentioned in the introduction. However, I would expect a clear background and terminology building in this section, i.e., authors should not only name, e.g., “code bots”, but should also explain what they are, which tasks they do, etc. Long story short, I consider the related work insufficient and suggest the authors to create the two subsections “background” and “related work”, such that the scene is set, the terminology is introduced, and the state of the art is properly presented. Finally, authors should discuss gaps in the field to provide an argument why the study is necessary (more on that in the study design section).

2) I apologize for the clear and hard statement, but the research design is poor. Everything about the research design seems to be located in section 3. This section needs to be completely revised. Details on this in the study design section of this review.

Experimental design

The study design is the weak point of this paper. In the submission-system abstract, the authors state to report a systematic literature review (SLR). In the actual paper, authors talk about a systematic mapping study (SMS). Unfortunately, the paper does not allow for judging which of these study types have been applied. The results section reads like an SLR. However, if the authors conducted a mapping study, key aspects are missing, notably:
- A classification schema
- A collection of maps
To make my point clear, SMS and SLR both have different targets. An SMS is aimed to explore a domain of interest and to provide an overview of the field by classifying the publications, while an SLR explicitly aims to generate new knowledge from analyzing a publication body. In the current version, neither of these goals is properly addressed.

Moreover, the research design is weak missing essential parts. Let’s start with the research questions. On page 2, l. 98, authors state “We surveyed the research articles pondering upon these three research questions” – however, even with a text search I was not able to find the research questions. This is the show stopper. Since I don’t know what the research questions are, I just cannot evaluate the paper properly and, therefore, I cannot provide much of a review of the findings (see respective section).

Furthermore, the research design lacks important information:
- Rationale for the study
- Search queries – the construction procedure/rationale is missing
- Search strategy: incomplete and inconsistent. Google Scholar is still subject to debate whether or not it is a proper (meta) search engine. An argument why “traditional” literature databases have been excluded from the search is, however missing. The search strategy is also inconsistent described since the snowballing is kind of located after the in-/exclusion, which is kind of “creative”. This decision was, unfortunately, not explained.
- Paper selection: while in-/exclusion criteria are listed, the actual paper selection procedure is not explained.
- The data extraction procedure is not explained.
- Validity procedures and the actual threats to validity are not discussed

Therefore, I suggest that the authors first clarify which kind of study the are reporting and, second, after having selected a study type, I suggest consulting the major references for executing and reporting these studies, i.e.:
- Kitchenham et al.: Evidence-Based Software Engineering and Systematic Reviews
- Petersen et al.: Systematic mapping studies in software engineering

I apologize again for my hard statement. Without this information, the article just cannot be accepted since the requirements for transparent and reproducible research are not met.

Validity of the findings

Well, I have to say the text in the results section is quite informative and, so, authors did a lot of work. However, since the research questions are missing, I just cannot seriously review this section. Since I just don’t know the exact questions, every review would be grounded in my personal interpretation. I kindly ask the authors for understanding that I cannot do this.

Additional comments

In general, the overall writing of the paper is good and I have no further comments to be added to my list in the basic reporting section.

My only general comment is that authors need to revise and probably rethink this article. It might happen that, after deciding on an actual study type, a considerable amount of rework is necessary, maybe including another data collection stage. Therefore, authors should critically discuss and revise their paper.

---

## Round 0.2 · accepted · Accept

I have verified the changes suggested by the reviewers and your rebuttal to these points. Good work with the updates. I am happy to accept this now.